# A Review of *Ganoderma lucidum* Polysaccharide: Preparations, Structures, Physicochemical Properties and Application

**DOI:** 10.3390/foods13172665

**Published:** 2024-08-24

**Authors:** Yuanbo Zhong, Pingping Tan, Huanglong Lin, Di Zhang, Xianrui Chen, Jie Pang, Ruojun Mu

**Affiliations:** College of Food Science, Fujian Agriculture and Forestry University, Fuzhou 350002, China; 19983416552@163.com (Y.Z.); ppingtan@163.com (P.T.); 15305943119@163.com (H.L.); zdfst@fafu.edu.cn (D.Z.); cxrui0526@gmail.com (X.C.); pang3721941@fafu.edu.cn (J.P.)

**Keywords:** *Ganoderma lucidum* polysaccharide, extraction, purification, structures, bioactivities, quantitative, qualitative

## Abstract

*Ganoderma lucidum* (GL) is a kind of edible fungus with various functions and a precious medicinal material with a long history. *Ganoderma lucidum* polysaccharide (GLP) is one of the main bioactive substances in GL, with anti-tumor, anti-oxidation, anti-cancer, and other biological activities. GLP is closely related to human health, and the research on GLP is getting deeper. This paper reviewed the extraction and purification methods of GLP, the relationship between structure and activity, and the qualitative and quantitative methods. This review provides solutions for the analysis and application of GLP. At the same time, some new methods for extraction, purification and analysis of GLP, the relationship between advanced structures and activity, and future applications of and research into GLP were emphasized. As a kind of bioactive macromolecule, GLP has unique functional properties. Through the comprehensive summary of the extraction, purification, and analysis of GLP and its future prospects, we hope that this review can provide valuable reference for the further study of GLP.

## 1. Introduction of *Ganoderma lucidum* Polysaccharide

*Ganoderma lucidum* (GL) is a group of fungi belongs to basidiomycete. It is widely distributed in tropical, subtropical, and temperate regions of Europe, America, Africa, and Asia. GL is mainly composed of ash (0.72%~1.77%), carbohydrates (21.83%~27.78%), fats (1.1%~8.3%), fibers (59%~65%), proteins (7%~8%), etc. [1]. It has been reported that there are several active constituents in GL, such as triterpenes, polysaccharides, steroids, fatty acids, amino acids, nucleosides, proteins, alkaloids, inorganic elements, etc. [2]. Most of the above constituents have been proven to have numerous health benefits due to their immunomodulatory effect [3,4], antioxidant [5], anti-tumor [6,7], anti-cancer [8,9,10,11,12], treatment of diabetes [13], prevention of metabolic syndrome cardiovascular disease [14], anti-obesity [15], and regulation of intestinal microorganisms properties [16]. These varied effects enhance the overall medicinal value of GL.

*Ganoderma lucidum* polysaccharide (GLP), one of the representatives of active ingredients in GL, is a substance extracted from GL spore powder or fruiting bodies [17]. As a natural biomacromolecule, GLP has been widely used in food, medicine, and health products due to its beneficial effects on human health (Figure 1). This promotes more intensive research on the structures, activities, and applications of GLPs. Most recently, OGLP-CMC/SA hydrogel was synthesized by using oxidized GLP, sodium alginate, and carboxymethyl chitosan as a matrix. This hydrogel has good mechanical properties, antioxidant properties, and biocompatibility, and it can effectively reduce inflammation and promote epidermal growth so as to heal diabetic wounds [18]. In another study, a dissolvable microneedle patch was synthesized using WSG, a polysaccharide extracted from GL, which inhibits the viability and mobility of melanoma cells, with two biocompatible compounds, PVA and PVP. This patch can significantly inhibit the growth of melanoma, breaking through the conventional treatment [19].

In the past few years, the importance of GLP has received many detailed reviews. For example, GLPs can be used as novel neuroprotective agents [20], as well as in biosynthesis [21] and clinical research [1]. All these indicate the importance of GLP research, but the latest research results of GLPs, such as the new methods of preparation and detection and the relationship between advanced structure and biological activity, have not been summarized. These new research contents can not only improve the shortcomings of traditional methods but also provide scientific direction for further research on GLP in the future. Therefore, this review will not only summarize the past valuable research but also summarize the latest cutting-edge research results and put forward the prospect of future research directions for GLP.

And as we carry out this work, we find there are still several unsolved problems related to the structures and functions of GLPs. Firstly, the structure of GLPs is diverse and complex, and most of the current research focuses on their primary structure. The conformations and three-dimensional (3D) structures of GLPs are still seldom investigated. Secondly, GLPs are mostly applied in drugs and health care products, but only several species are legal in the food industry. The National Health Commission and the State Administration for Market Regulation said, in a document released in 2023, that *G. lucidum karst* and *Ganoderma sinense* can be eaten as food and Chinese medicine. Further investigation on toxicology and food safety analysis of various species of GL may promote more applications of GLPs in food products. Thirdly, the contents of GLPs are varied depending on the species, original sources, and growth stages of GL. There is a trend towards developing real-time monitoring procedures and image systems to detect the content of GLPs. The most advanced machine-learning technology is potentially critical for the construction of databases related to bioactive ingredients in GL. In this paper, we will briefly summarize preparations, structure–activity relationships, and analysis technology of GLPs. At the same time, we will try to propose ideas that may solve the above-presented problems. We will investigate spectroscopic techniques applied to the characterization of the advanced structures of GLPs, such as near-infrared, hyperspectral, and mass spectrometry. Several new advanced and integrated methods are described, such as mass spectrometry with imaging systems and spectroscopic techniques with machine learning for rapid and nondestructive detection. It is hoped that we can provide novel ideas for future research on GLPs.

## 2. Preparation of *Ganoderma lucidum* Polysaccharide

### 2.1. Extraction of GLP

GLP mainly comes from the fruiting body and mycelium, and most of it exists in the cell wall. Therefore, in the process of extracting GLPs, it is necessary to consider how to effectively break the cell wall and prevent the degradation of GLPs to ensure their extraction rate. At present, there are four methods used to extract polysaccharides from GL, including physical methods, chemical methods, biological enzymatic methods [22], and mechanical crushing. The physical methods mainly include the water extraction method [23,24], ultrasonic assisted extraction method [25,26], and microwave method [27]. The chemical method is mainly the alkali extraction method [28]. The mechanical crushing method is the use of a twin-screw extruder for extrusion blasting [29]. Those methods have long been investigated and applied in the preparation of GLPs. There are many reviews that detail the specific steps and principles of these traditional methods [30,31,32]. Therefore, we will not explore each method in detail in this review. In order to improve the extraction efficiency of a single method, many researchers will combine two methods, such as ultrasound-assisted enzyme method [33], ultrasonic microwave-assisted method [34], and vacuum-microwave extraction [35]. This combination can indeed improve the extraction rate of GLP, but it greatly increases the extraction time.

Most recently, several new strategies have been created to extract GLPs. The first method is the fermentation method [36]. It uses the transformation of microorganisms (Bacillus and Saccharomyces cerevisiae) to extract bioactive substances more gently, which can help to retain the natural active ingredients in the extract and reduce the toxic side effects. At the same time, microorganisms will produce a variety of active enzymes (cellulase and protease) in the metabolic process to achieve synergistic effects. Compared with the above traditional methods, it has a higher utilization rate of raw materials and a milder action condition. The second method is ternary deep eutectic solvent extraction [37]. It is the synthesis of DESs by the molar ratio of choline chloride, guaiacol and lactic acid at 1:1:1. The key parameters were optimized by response surface method, and the maximum extraction rate was 94.72 g/mg. In this process, there is a triple hydrogen bond interaction and high binding energy between DESs and glucose, which is the main reason for improving the extraction rate. DESs has good cycle stability and high recovery rate, reducing the consumption of raw materials and environmental pollution [38]. The third method is continuous phase transition extraction [39]. The researchers first applied the technique to GLPs extraction. In the extraction process, continuous fresh solvent enhances the concentration gradient and increase the mass transfer rate, thus extracting more polysaccharides. Compared with the hot water method and ultrasonic method, the extraction rate of polysaccharide after CPTE was 3.34 times and 2.68 times. This method has the advantages of time-saving and high efficiency, so it is a promising extraction method.

Currently, GLPs are extracted by using traditional methods regularly, but it may lead to less polysaccharide and more impurities. For example, the water extraction method does not completely destroy the cell structure. Alkali extraction will use organic reagents that may cause pollution to the environment, and the type and concentration of alkali should be considered when using this method to avoid polysaccharide degradation. Although the effects of the ultrasonic-assisted method and microwave-assisted method are better than the first two, they have no selectivity in the extraction process and make it easy to produce more impurities. As a mild method, enzyme extraction has high extraction efficiency, but its price is high, and the conditions of enzyme action need to be strictly controlled to avoid its inactivation. The mechanical method is seldom used at present because its parameters are difficult to control. For these cutting-edge methods, although the extraction efficiency is better than that of traditional methods, they are in the embryonic stage and need more time to verify their feasibility. Different extraction methods have different effects, which may affect the variety, structure, and biological function of polysaccharides, so it is necessary to select the appropriate extraction method according to many factors. With the development of efficient green concepts, researchers can not only optimize on the basis of traditional methods but must also develop more new extraction techniques. These are the research focuses in polysaccharide extraction in the future. As for the advantages and disadvantages of the above-mentioned GLP extraction methods, we summarize them briefly in Table 1.

### 2.2. Separation and Purification of GLP 

Extraction procedures are essential to obtain GLPs, which are then still filled with various impurities like proteins, pigments, and other organic small molecules [48]. Therefore, further purification procedures are critical to obtain highly purified and structurally homogeneous GLPs. In this section, we will investigate in detail the several steps of the separation and purification of GLPs, as well as the commonly used methods.

The presence of proteins affects the quality of the polysaccharide and its physiological activity and may also influence the subsequent structural analysis of the polysaccharide, etc. Therefore, the extracted polysaccharides need to be deproteinized to obtain purified polysaccharides. One most commonly used method, called the “Sevag” method, is based on the interaction of organic solvents and salts to form a precipitate of proteins in solution for separation and purification [49]. Another method is that of using trichloroacetic acid as a denaturant to remove proteins from polysaccharides [50]. This method is based on a basic principle, in which proteins are desaturated under acidic conditions. While conformation of proteins is modified and insoluble salts are generated, more hydrophobic groups exposed those favorable for forming precipitates. However, the above two methods require the use of organic reagents, which may cause degradation of the polysaccharides [51], affecting the subsequent determination of their characterization of their structure. The third method is the protease method. In this, the polysaccharides are not mixed with organic reagents but rather enzymes to break the connecting bonds between polysaccharides and proteins, thus causing less contamination [52]. Moreover, after deproteinization with protease, the total carbohydrate recovery and the efficiency was high, and the antioxidant activity of GLPs could still be maintained at a high level [51]. The high cost of enzymes makes it difficult to realize industrial applications.

The pigment mainly comes from the polysaccharide itself and extraction process residues, and it will affect the detection results of the subsequent experiments, especially the color reaction of the experiments [48]. Meanwhile, formation of pigments is a major factor in the structural characterization and bioactive complexity of polysaccharides [53]. The most-used method of removing colorants is the activated carbon adsorption method. Activated carbon is an adsorbent material with a highly microporous structure, and organic and inorganic substances and other impurities in the object can be adsorbed on its surface to achieve purification. It captures pigments in polysaccharides mainly by van der Waals and electrostatic forces, but the selectivity coefficients for pigments and polysaccharides are poor [54]. Another method is to use hydrogen peroxide decolorization, which has a strong oxidation capacity and can break the double bonds in colored substances [55]. The third method is to use macroporous adsorption resins, which can adsorb pigments through their large surface area. In addition, macroporous resins have the advantages of good stability, low cost, and high adsorption efficiency. The resin is also regenerable, making it more suitable for polysaccharide decolorization than the previous two methods [56].

Deproteinization and decolorization are only for the removal of proteins and pigments. There are still many small molecules in polysaccharides, especially some inorganic salts, monosaccharides, and oligosaccharides. Ultrafiltration is a pressurized membrane separation technology in which small molecule impurities pass through a film with certain-sized pores under a pressure difference and polysaccharides are retained, thus realizing the purpose of separation and removal of impurities [57,58]. Dialysis is also a membrane separation method that uses diffusion pressure to expel small molecule impurities out of a semi-permeable membrane while large molecule polysaccharides remain in the membrane [59,60].

Graded precipitation [61] is a method that uses ethanol, methanol, and propanol as precipitants and precipitates the precipitate according to the different solubilities of polysaccharides in organic solvents of different concentrations. The second method is column chromatography, including anion exchange method, gel chromatography, and macroporous resin column chromatography. To obtain the homogeneous polysaccharide, GLPs can be purified by anion exchange combined with gel column [62]. The third method, ultrafiltration, is a pressure-driven membrane separation technique that removes small molecule solutes and solvents.

The above methods are commonly used for GLPs’ separation and purification, but their shortcomings could not be ignored. In deproteinization, the “Sevag” method needs to be repeated several times to achieve the desired results. Like trichloroacetic acid, the “Sevag” method leaves toxic chemical residues and may causes polysaccharide degradation. In the decolorization process, the selectivity coefficient of activated carbon for polysaccharide and pigment is poor [63], which makes it easy to cause the loss of polysaccharides. The strong oxidation power of hydrogen peroxide will cause the degradation of polysaccharide and reduce its molecular weight. Furthermore, the molecular weights of polysaccharides are different, and the resolution of ultrafiltration is low, so it is necessary to consider the molecular weights of polysaccharides and select different conditions. With the updating of technology, some new separation and purification techniques have appeared. For example, repeated freeze–thaw treatment causes insoluble aggregation and precipitation of proteins by causing complex changes in the buffering environment [64,65]. Dialdehyde cellulose is a modified polysaccharide that can be used to form a Schiff base by binding to proteins, thereby removing proteins from some crude polysaccharides [66]. Compared with traditional methods, these two methods have higher deproteinization efficiency and polysaccharide extraction rate, and the reagents and treatment processes used are greener and safer, avoiding environmental pollution. Besides these new methods of deproteinization, asymmetrical flow field-flow fractionation (AF4), ultrafiltration and the “Sevag” method are combined to improve the purification process of GLP [67]. AF4 provides sample component separation based only on its hydrodynamic size. The upper wall of the AF4 channel is an impermeable polycarbonate glass plate, and the bottom channel plate is permeable. The bottom channel plate and ultrafiltration membrane form an accumulation wall, which allows free passage of the carrier liquid and small molecules with sizes smaller than the MWCO. This method combines a variety of methods and uses deionized water throughout the research process, avoiding the process of dialysis desalination and greatly improving the efficiency while retaining the original structure of GLPs.

There is no specific standard for the extraction, separation, and purification of GLPs. Different treatment methods have different advantages and disadvantages, and the polysaccharides obtained are also different. GLP is an edible active substance, so the whole process needs to be green and safe as the first principle. Therefore, avoiding using harmful organic solvents or using less-toxic ones throughout the whole process is advisable to avoid subsequent products containing toxicity and limit their application. Secondly, efficiency should be considered. GLP has a broad application prospect in medicine and health food and will inevitably achieve large-scale industrial production with further development. When other conditions permit, efficient methods can be beneficial to economic returns. Therefore, exploring more green and efficient methods to obtain safe, large-quantity and high-purity GLPs is still the focus of future research.

## 3. Structure of *Ganoderma lucidum* Polysaccharide

### 3.1. Structure from Primary to Quaternary

It has been shown that structure of natural polysaccharides may be categorized into primary, secondary, tertiary, and quaternary structures [20]. The primary structure mainly includes the composition of the monosaccharides, the way in which neighboring sugar groups are connected, the allosteric configuration, and the presence or absence of branches [68]. The composition of monosaccharides of GLP from different sources may vary, with most of them mainly containing glucose and galactose and some of them also containing mannose, arabinose, and fucose [21]. These polysaccharide compounds can be categorized as homopolysaccharides and heteropolysaccharides according to the composition of the monosaccharides [3]. Polysaccharides can also be categorized into α-type and β-type based on the isotropic structure of monosaccharides. As the main active structure, β-type polysaccharides have received much attention from researchers [69]. However, there are few studies on the activity of α-type polysaccharides. Monosaccharides are linked to each other by different glycosidic bonds such as (1→3), (1→4), and (1→6). This is the main reason that most GLPs present high biological activities. 

The composition of monosaccharides, types of glycosidic bonds, and anisotropic structures described above are only based on primary structural analyses of GLPs. The term “advanced structures” primarily includes secondary, tertiary, and quaternary structures of GLPs. Secondary structures refer to various types of polymers with hydrogen bonding between chains of polysaccharides. At present, the secondary structures found in GLPs are single helical structures [70,71], rigid chain conformation [72], and linear and short-rod conformation [73]. Tertiary structures are further twisted and folded on the basis of the secondary structure format to form the spatial conformation with a certain shape and size. Researchers used HPSEC-MALLS-RI to identify a polysaccharide in GL with a compact sphere chain conformation, formed by stacked multiple chains [74]. Another study found that GLP is circular aggregates formed by intertwined chains and has a triple helix structure in aqueous solution [75]. The quaternary structures are the polymer formed by the combination of non-covalent bonding between macromolecular chains. Due to technical limitations, there is no research to clarify the quaternary structure type of GLPs, which is also one of the problems to be overcome in the future.

### 3.2. Structure with or without Activity

The structure of GLP is closely related to its biological activity. The main factors affecting the structure and bioactivity of GLP include monosaccharide composition, glycosidic bond, side chain composition, and molecular weight [76]. According to the composition of monosaccharides, polysaccharides can be divided into homopolysaccharides and heteropolysaccharides. Homopolysaccharides can significantly inhibit tumor growth and down-regulate proliferating cell nuclear antigen markers [77]. Heteropolysaccharides also show good anti-tumor, immune stimulation, and other activities because heteropolysaccharides are different monosaccharides combined in different proportions or ways. One of the key ingredients is mannose, which helps human organisms recognize toxic carbohydrates, thus stimulating the immune system to produce cytokines [78]. It has been reported that heteropolysaccharides exhibit higher immunomodulatory effects than homopolysaccharides [73]. Furthermore, the active polysaccharides isolated from GL are mainly β-(1→3), β-(1→4), and β-(1→6) types of glucans [79]. β-(1→3)-D-glucan is not only resistant to solid tumors [80] but also acts as an antimicrobial agent [79]. The low-molecular-weight β-(1→3)-glucan also reduces the formation of reactive oxygen and inhibits acidic and neutral sphingomyelinase activity [81]. Purified β-(1→3)-glucan also has an immunizing effect [72,82]. β-(1→6)-D-glucan has also been reported to have antitumor activity and activation of NKs [83]. Besides glucan, other monosaccharides can be linked by different glycosidic bonds, such as β-(1→3)-D-glucose [84], β-(1→3)-D-glucopyranosyl [7], β-(1→6)-D-glucopyranosyl [84], β-(1→3)-glucohexaose [85], etc., and exhibit different activities. Of course, there are α-types of polysaccharides that have been isolated, but due to their water-insoluble nature, their activity has been less studied. The isolation of α-(1→3)-D-glucan from GL substrates, a water-insoluble polysaccharide, was chemically modified by researchers to improve its water solubility. The anti-tumor [86], antioxidant [87], and immunomodulatory [88] activities of α-(1→3)-D-glucan were significantly enhanced after chemical modification. In addition to the differences in monosaccharide composition and glycoside bond, the molecular weight also affects the activity of polysaccharides. In one study, a homopolysaccharide with a molecular weight of 44.4 kDa was isolated from the fruiting body of GL and showed good antitumor activity in mice [77]. Another study isolated a homopolysaccharide with a molecular weight of 1013 kDa, which also has anti-tumor effects [89]. From these two studies, it can be seen that no matter the size of the molecular weight of polysaccharide, it has a certain biological activity. However, one report suggested that the greater the molecular weight of GLP, the higher its biological activity [90]. Therefore, there is still disagreement about which molecular weight has the best biological activity, and further research is needed. We summarize information about the primary structure of some GLPs with respect to their biological activities and other aspects in Table 2.

In addition to the primary structure being related to the activity of GLPs, there is a definite link between its advanced structure and activity [91]. The triple helix structure is the most common type of advanced structure of polysaccharides and the most studied [92]. Researchers demonstrated that GLP with triple-helical structure can modulate immune activity through RAW264.7 macrophage in a cell model study in vitro [75]. There are also some GLPs that show linear or short rod conformation, while the short rod conformation has more immunomodulatory activity than the linear one [73]. This may be due to the different compositional ratio of glucose in the formation of polymers. GLP can also form a rigid chain conformation in aqueous solution [72]. The polysaccharide changes conformation from an ordered structure to a single-chain structure when the NaOH concentration is greater than 0.15 M in an alkaline solution or in an aqueous solution at 135 °C or above. This transition is consistent with experimental results with other edible mushroom polysaccharides [93], and it is hypothesized that the biological activity of this polysaccharide is related to the rigid chain conformation. However, this conformational shift may result in a decrease or loss of its activity [94]. Because of the limited technology and the hindrance of the complex structure of polysaccharides, the relationship between advanced structure and biological activity has been less studied, and this is one of the challenges to be overcome in the future.

**Table 2 foods-13-02665-t002:** The structures and bioactivities of GLPs.

Type	Backbone	Name	Mw	Monosaccharide Composition Ratio	Bioactivity	Raw Source	Reference
α	α-(1→6)-D-galactopyranosyl α-(1,2,6)-D-galactopyranosyl	LZ-D-1	2.8 × 10^4^ Da	L-Fuc:D-Glc:D-Gal = 1:1:5	Immunity: Stimulate proliferation of mouse spleen lymphocytes in vitro	Chongming/ Shanghai	[95]
	α-(1→4)-D-glucan	LB-B1	9.3 × 10^3^ Da	Only D-glucose	—	—	[96]
	α-(1,6)-Galp	PSG-2	6.9 × 10^4^ Da	Galactose:Fucose:Glucose = 8:1:1	—	Ganzhou/ Jiangxi	[97]
β	β-(1→3)-D-glucan	GL-IV-I	1.33 × 10^5^ Da	—	—	Longyan/Fujian	[71]
		GLP20	3.75 × 10^6^ Da	—	Immunity: Increase NO production of RAW264.7 macrophages	Shanghai	[72]
		PSGL-I-1A	7.18 × 10^5^ Da	Only D-glucose	Immunity: Affect T lymphocyte-stimulating activity	Shanxi	[98]
	β- (1→3)-D-glucosyl	LB-NB	4.7 × 10^4^ Da	Only D-glucose	Immunity: Remarkable stimulation of proliferation of T-cells in vitro	Shanghai	[70]
	β-(1→3)-glucose	PSG-1	—	Glu:Mannose:Galactose = 9:1:1	—	—	[99]
	β-(1→3)-D-glucopyranosyl	SP	1.0 × 10^4^ Da	—	Immunity: Enhancement of lymphocyte proliferation and antibody production	—	[100]
	β-(1→6)-D-glucan	PGL	1.26 × 10^5^ Da	Only D-glucose	Immunity: Had an immunosuppressive effect on antibody production and lymphocyte proliferation	—	[101]
	β-(1→3)-(1→6)-D-glucan	GTM5	1.76 × 10^6^ Da	Glc	Antitumor	Wuhan	[80]
		GTM6	1.61 × 10^6^ Da	Glc:Man = 3.83:1	Antitumor	Wuhan	[80]
	β-(1→3)-(1→4)-(1→6)-D-glucopyranosyl β-(1→6)-D-mannopyranosyl	PL-4	2.0 × 10^5^ Da	Mannose:Glc = 1:13	Immunity: Enhanced the proliferation of T- and B-lymphocytes in vitro	Shanxi	[102]
	β-(1→3)-(1→6)-D-glucopyranosyl	Ganoderans B	7.4 × 10^3^ Da	—	Reduced the blood glucose concentration	Kyoto/Japan	[103]
	β-(1→3)-(1→4)-(1→6)-D-glucan	GLSWA-I	1.57 × 10^5^ Da	—	Significantly promoted dinitrochlorobenzene-induced delayed-type ear swelling in mice	Shanghai	[104]
	β-(1,6)-D-Glcp	GLSA50-1B	1.03 × 10^5^ Da	Only Glucose	—	Shanghai	[105]
α, β	α-(1→4)-D-glucan β-(1→3)-D-glucan	GTM3	4.65 × 10^6^ Da	Glc	Antitumor: Exhibited significant inhibition ratio beyond 50%	Wuhan	[80]
		GTM4	4.68 × 10^6^ Da	Glc	Antitumor	Wuhan	[80]
	α-(1→4)-D-glucopyranosyl β-(1→6)-D-galactopyranosyl	PL-1	8.3 × 10^3^ Da	Rha:Gal:Glc = 1:4:13	Immunity: Enhanced the proliferation of T- and B-lymphocytes in vitro	Shanxi	[102]
	α-(1→6)-D-galactopyranosyl β-(1→3)-(1→6)-D-glucopyranosyl	Ganoderans C	5.8 × 10^3^ Da	D-glucose:D-galactose = 24:1	Reduced the blood glucose concentration	Kyoto/Japan	[103]
	α-(1,6)-galactopyranosyl α-(1,2,6)-galactopyranosyl β-(1,3)-glucopyranosyl β-(1,4,6)-glucopyranosyl	LZ-C-1	7.0 × 10^3^ Da	L-Fuc, D-Glc, D-Gal	—	Shanghai	[106]
	α-(1→6)-D-glucopyranosyl β-(1→3)-D-glucopyranosyl β-(1→3,6)-D-glucopyranosyl	GSG	1.43 × 10^5^ Da	—	Immunity: Stimulating effects on murine lymphocyte proliferation	Jilin	[107]
Not well-known	1,2,6-galactose 1,3-glucose 1,6-galactose	GLPCW-II	1.2 × 10^4^ Da	D-Glc:L-Fuc:D-Gal = 1.00:1.09:4.09	Stimulated the proliferation of mouse spleen lymphocytes	Shanghai	[108]
	—	PSG-1	1.013 × 10^6^ Da	Glucose:Mannose:Galactose = 4.91:1:1.28	—	Ganzhou/ Jiangxi	[97]
	1,3-glucosyl	GSG	8.0 × 10^3^ Da	Only D-glucose	Immunity: Potentiated the Con A-induced proliferative response of splenocytes	—	[109]
	—	GLIS	—	D-glucose:D-galactose:D-mannose = 3.0:1:1	Immunity	—	[110]
	—	SeGLP-2B-1	1.06 × 10^6^ Da	Glucose:Rhamnose:Xylose:Galactose = 1.000:0.652:0.443:0.227	Anticancer	—	[111]
	Glucan	PL-3	6.3 × 10^4^ Da	Glucan	Immunity: Enhanced the proliferation of T- and B-lymphocytes in vitro	Shanxi	[102]
	—	GTM1	6.28 × 10^5^ Da	Galactose:Mannose = 1.85:1	Antitumor	Wuhan	[80]
	—	GTM2	8.18 × 10^5^ Da	Galactose:Glc = 1:1.36	Antitumor	Wuhan	[80]

## 4. Analysis of *Ganoderma lucidum* Polysaccharide

Polysaccharide is an important active substance in GL, which has many biological activities and can promote human health. The content of polysaccharides in GL has always been one of the key concerns of consumers, and its content will change with the growth region and growth stage of GL. How to achieve non-destructive, rapid, and real-time detection of polysaccharide content is a future development trend. The structure of polysaccharides is complex and diverse. Different monosaccharides and different glycosidic bonds can form a variety of primary structures, and different primary structures can establish different advanced structures and conformations. So, figuring out its structure is a challenge. In this section, we will review the qualitative and quantitative methods of GLPs, with emphasis on some novel methods.

### 4.1. Quantitative Analysis of GLP

We summarize all appliable analysis methods for quantifying GLP in Figure 2. At present, the traditional methods used for the detection of GLPs content mainly include the phenol sulfuric acid method [112,113] and anthrone sulfuric acid method [114], both of which utilize the principle of chromogenic reaction for detection. Their results are susceptible to various factors, and they require complex pre-treatment and the use of organic reagents that can easily cause environmental pollution [115]. HPLC [116] is beginning to be applied for detecting the content of GLPs because of its high sensitivity and accuracy. High-performance size exclusion chromatography (HPSEC) is a type of HPLC and is a good method for quantitative analysis. The polysaccharide content in GL can be accurately analyzed by combining it with multi-angle laser scattering [117]. However, it requires a large amount of expensive organic substances and laborious pre-treatment. The determination of GLPs content using the above method is quite time-consuming and inefficient when the sample amount is too large. In order to realize non-destructive and rapid detection, it is essential to find some novel and rapid detection methods.

Machine learning is an increasingly attractive technology that is widely applied in various areas, such as medicine [118,119], materials chemistry [120,121], and biology [122]. Most recently, several reviews [123,124] investigated the potential application of machine learning in food science and industry. It is obvious that this advanced technology will help to solve many scientific and technological problems during research on food topics. Detection of food constituents is a sophisticated procedure with a series unit process. As we discussed above, the traditional method requires the sample to be crushed and then extracted, separated, and purified before quantitative analysis, which is very complicated and cumbersome. Therefore, machine learning has been combined with many technologies, such as hyperspectral imaging [125] and infrared spectroscopy (mid-infrared spectroscopy [126] and near-infrared spectroscopy [127]), to analyze GLPs with high efficiency, fast data analysis, and little or no sample preparation, followed by machine learning to extract data features to build models. This cutting-edge approach enables real-time monitoring of polysaccharide content in GL regardless of species, region, and growth stage and provides a viable method for improving the quality and economic value of GL, with promising applications in large-scale cultivation and high-throughput detection.

**Figure 2 foods-13-02665-f002:**
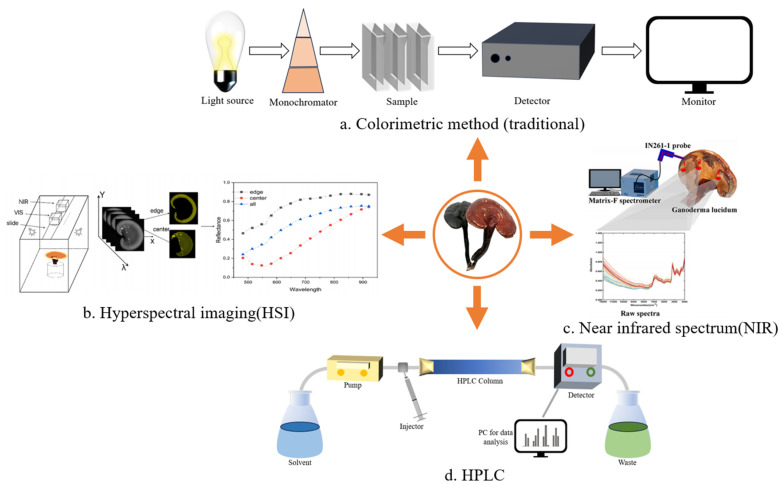
Methods for quantitative analysis of GLP. (**a**) The content of polysaccharide was calculated by measuring the absorbance of the sample using colorimetric reaction. (**b**) In a dark environment, the samples are photographed by halogen lamp and hyperspectral camera, and then the relationship between reflectance and wavelength was obtained by software processing and the polysaccharide content was finally calculated. (**c**) The infrared spectrum of GL was collected by probe, the performing band was selected by synergy interval partial least squares, and the model was optimized by ant lion optimization algorithm to predict the total polysaccharide content. (**d**) The chromatogram was obtained by HPLC, and the content of GLPs was deduced by calculating the peak area. (**b**,**c**) are reproduced from [125,127,128,129].

### 4.2. Qualitative Analysis of GLP

The methods for characterization of the primary structure of GLPs have been well established (Figure 3). Characterization of the structure of GLPs is mainly based on two strategies, chemical and instrumental. Commonly, chemical analysis methods include hydrolysis process, periodate oxidation, “Smith” degradation, and methylation analysis. The hydrolysis method includes complete acid hydrolysis, partial acid hydrolysis, etc., which are mainly used to analyze the composition of polysaccharide chains. Acid hydrolysis may lead to excessive degradation of monosaccharides under harsh conditions or prolonged treatment, so it is essential to explore the optimal hydrolysis conditions for polysaccharides [130]. Periodate oxidation is a common method for analyzing the structure of polysaccharides, and the position and type of glycosidic bond can be determined according to the consumption of periodic acid [128]. The methylation analysis mainly characterizes the linking mode of sugar residues but can degrade polysaccharides. Some studies have achieved complete methylation by improving the scheme, without causing obvious degradation [131]. Instrumental analytical methods mainly include GC, HPLC, HPAEC, MS, and NMR. Compared with chemical methods, this kind of method is more accurate and convenient [129,132]. Researchers have characterized GLPs through a variety of instrumental methods and elucidated the fine structural characteristics of its main chain and branch chain [99]. Another study analyzed the composition of *Panax species* polysaccharides by GC-MS as Rha, Ara, GalA, Man, Glc, and Gal [133]. The composition of monosaccharides, the composition of main and branched chains, and the determination of the type of glycosidic bond can be obtained by the above methods. Fewer methods are used to characterize advanced structures, mainly Congo red staining [134], light scattering, differential scanning calorimetry (DSC) [72], atomic force microscopy (AFM), circular dichroism (CD), and fluorescence spectroscopy. Congo red staining is the most convenient method to study the change of polysaccharide morphological chain, and it does not require significant equipment. CD is an effective method to study the three-dimensional structure of biological macromolecules, which can provide information about the absolute configuration and conformation of molecules [135]. The light scattering method can accurately reflect the change of chain conformation, but it requires high purity of the sample and solvent used. AFM only needs a small number of samples to analyze the morphology of polysaccharide chains, but it has high requirements for instrument equipment. At present, the methods used to determine the conformation of polysaccharides still have the problems of being time-consuming, high-cost and low-accuracy. It is a popular strategy to combine two different methods and equipment to study the conformation of polysaccharides [136]. The development of combining computer-aided and analytical methods for molecular modeling has begun [137]. This will provide a new more accurate and simpler strategy for the resolution of advanced structures of polysaccharides.

## 5. Conclusions and Future Perspectives 

GLPs have many biological activities and have positive effects on human health, so they have become a hot research topic at home and abroad. This article summarized the extraction and purification methods of GLPs, the relationship between structure and activity, and the qualitative and quantitative analysis methods of GLPs, but the research on GLPs needs to be further in-depth. According to the problems we found in the process of review, such as the complex structure of GLPs, the content being changeable, and the applications being too few, the following suggestions are put forward, hoping to provide useful value for future research on GLPs.

Some GLPs are insoluble in water, so it is difficult to satisfy all the chemical properties as well as exhibit satisfactory biological activity [17]. In order to solve this problem, their physical properties and chemical structure can be changed by chemical modification [138,139]. Nanoparticles prepared from natural polysaccharides have a special structure that enables the active ingredient to be encapsulated in a polymer matrix and precisely transported to a specific site for releasing [17]. GLPs can also be combined with other materials to form composite materials to improve their biological activity [140,141]. The study of chemical modification and nano delivery systems can make some GLPs show more obvious biological activity and increase their applications.The structure and activity of polysaccharides are closely related [142,143], but most of the current studies are on the relationship between primary structures and biological activity, and there are few studies on advanced structures and conformations. In future studies, we need to characterize these advanced structures and conformations with more novel methods and advanced instruments and elucidate their relationship with activity. This will provide us with more information about the biology of GLPs and scientific basis for their potential applications.As the main active substance of GL, polysaccharide has attracted more and more attention due to its remarkable biological activity. However, the scarcity of natural resources, the restriction of growth conditions, and the difficulty of controlling the stability of yield hinder its development and application. It would be a good solution to obtain GL strains with high polysaccharide content by artificial breeding with gene technology [144,145,146]. Of course, it is also possible to cultivate disease-resistant and pest-resistant GL through these technologies to improve the overall quality of GL.GL has different species, and their growth environments are not the same, which causes great differences in the content, efficacy, and structure of their bioactive ingredients. In order to facilitate researchers to understand information more quickly and clearly, it is essential to establish a shared database. This database can collect experimental data on GLPs from the whole world, and visitors can browse these data to explore the principles and mechanisms behind it and further promote the vigorous development of GL research.

## Figures and Tables

**Figure 1 foods-13-02665-f001:**
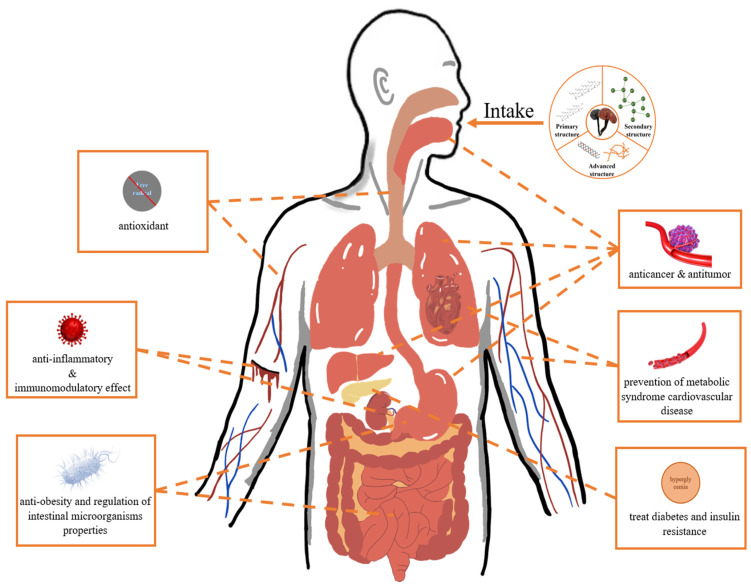
Relationship between activity of GLP and human health. GLP can protect human health through a variety of mechanisms, such as inhibiting the growth and proliferation of tumor cells, reducing activity of various enzymes, improving human blood pressure and cholesterol levels, and eliminating free radicals.

**Figure 3 foods-13-02665-f003:**
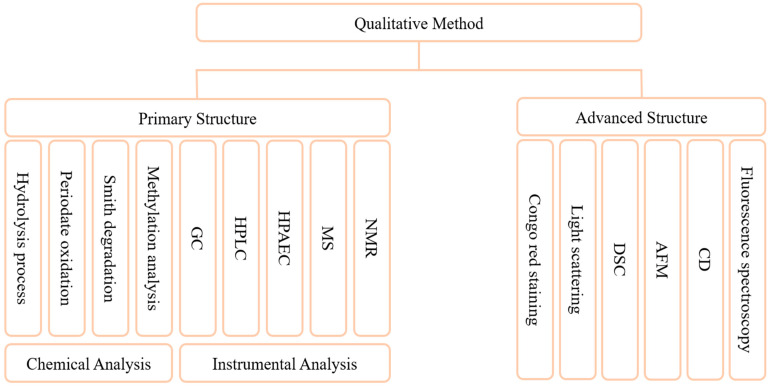
Qualitative methods of GLPs. These methods are used to analyze the composition of monosaccharides, the structure of main chain and branch chain, the judgment of the type of glycosidic bond, and the advanced structures.

**Table 1 foods-13-02665-t001:** The extraction methods of GLP.

Method	Technical Principle	Advantage and Disadvantage	References
Water extraction	Polysaccharides are soluble in water but insoluble in organic solvents	Advantage: Simple and safe, low-cost, will not cause polysaccharide degradation	[40]
		Disadvantage: Time-consuming, low extraction rate	
Alkali extraction	Under alkaline conditions, the fibers of GL decompose and accelerate the release of polysaccharides	Advantage: Efficient	[41]
		Disadvantage: Environmental pollution	
Ultrasonic-assisted extraction	Cavitation and mechanical effects	Advantage: Simple, high extraction rate, no material loss	[26,42,43]
		Disadvantage: Will destroy secondary and tertiary structures	
Microwave-assisted extraction	Using heat to rupture the cell wall	Advantage: Simple, efficient, no pollution	[31,44]
		Disadvantage: Excessive time can lead to degradation	
Enzyme extraction	Macromolecular substances are separated from GLP by enzymes, usually using complex enzymes	Advantage: Mild, efficient, high biological activity	[45]
		Disadvantage: Extraction efficiency is affected by enzyme activity, high cost	
Squeeze blasting	Set mixing, stirring, crushing, heating, blasting, sterilization, and molding as one of the high-tech methods	Advantage: High efficiency, low cost	[46,47]
		Disadvantage: The experimental parameters are difficult to control	
Fermentation extraction	Microorganisms produce active enzymes during fermentation	Advantage: High utilization of raw materials, mild	[36]
DESs	Triple hydrogen bond interaction and a high binding energy	Advantage: Commendable cyclic stability, high recovery rate, efficient	[37]
CPTE	Continuous fresh solvents enhance the concentration gradient and increase the mass transfer rate	Advantage: Fast, efficient	[39]

## Data Availability

No new date was created or analyzed in this study. Data sharing is not applicable to this article.

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
