# Peer review of "A Review of Ganoderma lucidum Polysaccharide: Preparations, Structures, Physicochemical Properties and Application"

_foods, 2024, doi:10.3390/foods13172665_

Round 1

Reviewer 1 Report

Comments and Suggestions for Authors

The review is an important and useful guide on how optimally proceed with Ganoderma Lucidum analysis and similar wood decay mushrooms.

Author Response

Dear Reviewer

Thank you very much for your hard work in processing our manuscript (foods-3135706R1) and for your recognition and support of our review.

Thank you and best regards.

Sincerely,

Ruojun Mu

College of Food science

Fujian Agriculture and Forestry University

Shangxiadian Road 15#, Cangshan District, Fuzhou, 350002

Reviewer 2 Report

Comments and Suggestions for Authors

The paper's topic is relevant and the perspective could be valuable for those interested. However, the manuscript needs significant improvement, particularly in its critical analysis. Here are a few comments:

In the introduction section, it is essential to include a more thorough discussion of the reach and impact of other reviews on the same topic. Incorporating and critically examining other authors' points of view could greatly enhance the scientific merit of this work.

Currently, the text lacks a critical analysis of existing literature. A review should not only summarize existing research but also analyze and synthesize the information to provide new insights and advancements in the field. Unfortunately, this article falls short in this aspect. The structured sections need more depth and critical evaluation necessary for scientific advancement. For example, it would be beneficial to include other perspectives, such as the influence of enzymes during plant biosynthesis, to provide a more comprehensive and critical view.

Overall, while the manuscript describes relevant works on Ganoderma, it fails to offer a critical analysis of the references it discusses. The review does not present a new perspective on applying current knowledge in technological applications, which is crucial for advancing the field.

In general, reviews must critically evaluate previous work and contribute new insights into the topic. As it stands, the manuscript does not clearly demonstrate the advances and their potential applications. The intention and delivery of the message are unclear, making it difficult to understand the paper's contribution to the field. Critical analysis and clearer articulation of the advancements and applications are necessary for this manuscript to meet the standards of a scientific review.

Author Response

Dear Reviewer

Thank you very much for your hard work in processing our manuscript (foods-3135706R1) and giving us an opportunity of the revise submission. We have added more results and discussions to address your comments in detail in the revised manuscript. After carefully studying the comments and your advice, we have made corresponding changes to the paper. All modifications are marked in red.

Responses to Reviewer #1

Comments: In the introduction section, it is essential to include a more thorough discussion of the reach and impact of other reviews on the same topic. Incorporating and critically examining other authors' points of view could greatly enhance the scientific merit of this work.

Currently, the text lacks a critical analysis of existing literature. A review should not only summarize existing research but also analyze and synthesize the information to provide new insights and advancements in the field. Unfortunately, this article falls short in this aspect. The structured sections need more depth and critical evaluation necessary for scientific advancement. For example, it would be beneficial to include other perspectives, such as the influence of enzymes during plant biosynthesis, to provide a more comprehensive and critical view.

Overall, while the manuscript describes relevant works on Ganoderma, it fails to offer a critical analysis of the references it discusses. The review does not present a new perspective on applying current knowledge in technological applications, which is crucial for advancing the field.

In general, reviews must critically evaluate previous work and contribute new insights into the topic. As it stands, the manuscript does not clearly demonstrate the advances and their potential applications. The intention and delivery of the message are unclear, making it difficult to understand the paper's contribution to the field. Critical analysis and clearer articulation of the advancements and applications are necessary for this manuscript to meet the standards of a scientific review.

Response: Thank you very much for your valuable comment. We have renewed or added some content according to the reviewer's comments. In the Introduction part, the views of other studies are added (P2 line47-56). And in the article, the views of the content of the review are also analyzed in depth (P3-P4 line116-135, P7 line224-234). These revisions are highlighted in red in the manuscript.

Line47-56: In the past few years, the importance of GLP has received many detailed reviews. For example, GLPs can be used as novel neuroprotective agents, as well as in biosynthesis and clinical research. All these indicate the importance of GLPs’ research, but the latest research results of GLPs, such as the new methods of preparation and detection, the relationship between advanced structure and biological activity, have not been summarized. These new research contents can not only improve the shortcomings of traditional methods, but also provide scientific direction for further research of GLP in the future. Therefore, this review will not only summarize the past valuable research, but also summarize the latest cutting-edge research results, and put forward the prospect of the future research direction of GLP.

Line116-135: Currently, GLPs are extracted by using traditional methods regularly, but it may lead to less polysaccharide and more impurities. For example, water extraction method does not completely destroy cell structure. Alkali extraction will use organic reagents may cause pollution to the environment, and the type and concentration of alkali should be considered when using this method to avoid polysaccharide degradation. Although the effect of ultrasonic-assisted method and microwave-assisted method is better than the first two, they have no selectivity in the extraction process and are easy to produce more impurities. As a mild method, enzyme extraction has high extraction efficiency, but its price is high, and the conditions of enzyme action need to be strictly controlled to avoid its inactivation. Mechanical method is seldom used at present because its parameters are difficult to control. For these cutting-edge methods, although the extraction efficiency is better than that of traditional methods, they are in the embryonic stage and need more time to verify the feasibility of these methods. Different extraction methods have different effects, which may affect the variety, structure and biological function of polysaccharides, so it is necessary to select the appropriate extraction method according to many factors. With the development of efficient green concepts, researchers can not only optimize on the basis of traditional methods, but also develop more new extraction techniques. Those are the research focus in the polysaccharide extraction in the future. As for the advantages and disadvantages of the above mentioned GLP extraction methods, we summarize them briefly in Table 1.

Line224-234: There is no specific standard for the extraction, separation and purification of GLPs. Different treatment methods have different advantages and disadvantages, and the polysaccharides obtained are also different. GLP is an edible active substance, so the whole process needs to be green and safe as the first principle. Do not use or use less toxic and harmful organic solvents in the whole process to avoid subsequent products containing toxicity and limit its application. Secondly, efficiency should be considered. GLP has a broad application prospect in medicine and health food, and will inevitably achieve large-scale industrial production with the development. When other conditions permit, efficient methods can be beneficial to economic returns. Therefore, exploring more green and efficient methods to obtain safe, large quantities and high purity GLPs is still the focus of future research.

Thank you again for all your suggestions. We hope that all these changes fulfill the requirements to make the manuscript acceptable for publication in foods and I am looking forward to hearing from you soon.

Thank you and best regards.

Sincerely,

Ruojun Mu

College of Food science

Fujian Agriculture and Forestry University

Shangxiadian Road 15#, Cangshan District, Fuzhou, 350002

Reviewer 3 Report

Comments and Suggestions for Authors

The paper entitled “A review of Ganoderma Lucidum polysaccharide: Structures, 2 Physicochemical Properties, and Application” is very interesting research; however, it is necessary to adjust throughout the text:

1.       The introduction is very long and does not contain information that is aligned with the title. should reflect on the initial content since the first 6 pages are not aligned to the title of the document. You should focus on You should focus on the main and most important ideas found in the title.

2.       From section 3 begins the information that is related to the title Line 233. Why do you claim that there was a physiological adaptation, what was the product or nutrient that you monitored?

3.       The references are mostly current, I only recommend you update  the ones that are from 2004 to 2013, most of them do not allow the article to be visualized as a current topic (frontier of science).

4.       Figure 2 is necessary, it only presents written information, I believe it should be omitted.

5.       The information presented on structure, physicochemical properties and applications is very interesting, however, it should emphasize and clearly state the relationship between structure and function. This comment is very relevant since it should reflect the importance of the review, otherwise it would look like a summary of reported papers.

6.       You should also make it clear why you are doing the review on GL, why not another? or others?

7.       You refer to the difficulty of recovering the exopolysaccharides due to its location is why separation methods are required using solvents etc. I suggest you indicate what are the real possibilities of scaling up for an industrial process as the information presented seems to represent a risk to the environment. You should indicate apart from the enzymatic method a practical method, or green method and to talk about the recovery of exopolysaccharides?

Author Response

Dear Reviewer

Thank you very much for your hard work in processing our manuscript (foods-3135706R1) and giving us an opportunity of the revise submission. We have added more results and discussions to address your comments in detail in the revised manuscript. After carefully studying the comments and your advice, we have made corresponding changes to the paper. All modifications are marked in red.

Responses to Reviewer #1

Comments 1: The introduction is very long and does not contain information that is aligned with the title. should reflect on the initial content since the first 6 pages are not aligned to the title of the document. You should focus on You should focus on the main and most important ideas found in the title.

Response: Thank you very much for your valuable comment. We have modified and condensed the content of the introduction and optimized the Section 2, and added “Preparation” to the title, hoping that this can be more consistent.

Comments 2: From section 3 begins the information that is related to the title Line 233. Why do you claim that there was a physiological adaptation, what was the product or nutrient that you monitored?

Response: Thank you very much for your valuable comment. This section mainly describes the structure of GLP, including primary structure and advanced structure. These structures are related to the biological activities of GLP, such as immune regulation and anti-tumor. This structural and functional information is also summarized in Table 2.

Comments 3:  The references are mostly current, I only recommend you update the ones that are from 2004 to 2013, most of them do not allow the article to be visualized as a current topic (frontier of science).

Response: Thank you very much for your valuable comment. We have updated the literature that can be updated and marked its citation in the paper in red.

Comments 4: Figure 2 is necessary, it only presents written information, I believe it should be omitted.

Response: Thank you very much for your valuable comment. We have removed Figure 2 from the article because this information can also be found in the text.

Comments 5: The information presented on structure, physicochemical properties and applications is very interesting, however, it should emphasize and clearly state the relationship between structure and function. This comment is very relevant since it should reflect the importance of the review, otherwise it would look like a summary of reported papers.

Response: Thank you very much for your valuable comment. The relationship between structure and active function is further described and highlighted in the paper (P8 line267-278, 292-301).

Line 267-278: The structure of GLP is closely related to its biological activity. The main factors affecting the structure and bioactivity of GLP include monosaccharide composition, glycosidic bond, side chain composition and molecular weight. According to the composition of monosaccharides, polysaccharides can be divided into homopolysaccharides and heteropolysaccharides. Homopolysaccharides can significantly inhibit tumor growth and down-regulate proliferating cell nuclear antigen markers. Heteropolysaccharides also show good anti-tumor, immune stimulation and other activities, because heteropolysaccharides are different monosaccharides combined in different proportions or ways. One of the key ingredients is mannose, which helps human organisms recognize toxic carbohydrates, thus stimulating the immune system to produce cytokines. It has been reported that heteropolysaccharides exhibit higher immunomodulatory effects than homopolysaccharides.

Line 292-301: In addition to the difference of monosaccharide composition and glycoside bond, the molecular weight also affects the activity of polysaccharide. In one study, a homopolysaccharide with a molecular weight of 44.4kDa was isolated from the fruit body of GL and showed good antitumor activity in mice. Another study isolated a homopolysaccharide with a molecular weight of 1013kDa, which also has anti-tumor effects. From these two studies, no matter the size of the molecular weight of polysaccharide, it has a certain biological activity. However, one report suggested that the greater the molecular weight of GLP, the higher its biological activity. Therefore, there is still disagreement about which molecular weight has the best biological activity, and further research is needed.

Comments 6: You should also make it clear why you are doing the review on GL, why not another? or others?

Response: Thank you very much for your valuable comment. GL is a precious Chinese medicinal material, which has a variety of biological activities and can promote human health. As a natural health product, its research can not only promote the development of traditional Chinese medicine industry, but also drive the development of food processing, agriculture, biotechnology and other related industries. There is also a brief introduction to GL in the first paragraph of Introduction. Natural polysaccharides have always been the focus of research in the field of biomedicine, food and health products. As one of the main active substances in GL, GLP has also been widely studied for its safety and non-toxicity. The in-depth study of GLPs can provide a more scientific basis for its application and better develop the market of GL. Therefore, this article takes GL as the object to review the polysaccharides contained in it, hoping to provide an important basis for the food development of GLP.

Comments 7: You refer to the difficulty of recovering the exopolysaccharides due to its location is why separation methods are required using solvents etc. I suggest you indicate what are the real possibilities of scaling up for an industrial process as the information presented seems to represent a risk to the environment. You should indicate apart from the enzymatic method a practical method, or green method and to talk about the recovery of exopolysaccharides?

Response: Thank you very much for your valuable comment. The main sources of GLPs are the fruit body and mycelium, and most of them exist in the cell wall. The extraction methods mentioned in this paper are aimed at destroying the cell wall and obtaining intracellular polysaccharides. Both intracellular and extracellular polysaccharides need to be separated and purified by the method mentioned in the article to obtain pure GLP. Enzymatic method, as a mild and non-toxic method, is suitable for polysaccharide extraction. However, the cost of enzymes is high and it is difficult to use them in industrial production. In the Section 2 (P3 line96-115), several new extraction methods are also proposed, but these methods are still in the new development stage, it needs to spend some practice to prove their practicability.

Line 96-115: Most recently, several new strategies are created to extract GLPs. The first method is fermentation method. It uses the transformation of microorganisms (Bacillus and Saccharomyces cerevisiae) to extract bioactive substances more gently, which can greatly retain the natural active ingredients in the extract and reduce the toxic side effects. At the same time, microorganisms will produce a variety of active enzymes (Cellulase and Protease) in the metabolic process to achieve synergistic effects. Compared with the above traditional methods, it has a higher utilization rate of raw materials and a milder action condition. The second method is ternary deep eutectic solvent extraction. It is the synthesis of DESs by the molar ratio of choline chloride, guaiacol and lactic acid at 1:1:1. The key parameters were optimized by response surface method, and the maximum extraction rate was 94.72g/mg. In this process, there is a triple hydrogen bond interaction and high binding energy between DESs and glucose, which is the main reason for improving the extraction rate. DESs has good cycle stability, high recovery rate, reducing the consumption of raw materials and environmental pollution. The third method is continuous phase transition extraction. The researchers first applied the technique to GLPs extraction. In the extraction process, continuous fresh solvent enhances the concentration gradient and increase the mass transfer rate, thus extracting more polysaccharides. Compared with hot water method and ultrasonic method, the extraction rate of polysaccharide after CPTE was 3.34 times and 2.68 times. This method has the advantages of time-saving and high efficiency, so it is a promising extraction method.

Thank you again for all your suggestions. We hope that all these changes fulfill the requirements to make the manuscript acceptable for publication in foods and I am looking forward to hearing from you soon.

Thank you and best regards.

Sincerely,

Ruojun Mu

College of Food science

Fujian Agriculture and Forestry University

Shangxiadian Road 15#, Cangshan District, Fuzhou, 350002

Round 2

Reviewer 2 Report

Comments and Suggestions for Authors

The manuscript is of adequate quality, the data is relevant and could be of interest to readers, I did not find  flaws in the methodology, and the discussion section is sufficient and in accordance with the manuscript objectives. In my opinion, the manuscript is ready to be published.